# An Increase in Fat Mass Index Predicts a Deterioration of Running Speed

**DOI:** 10.3390/nu11030701

**Published:** 2019-03-25

**Authors:** Laurence Genton, Julie Mareschal, Véronique L. Karsegard, Najate Achamrah, Marta Delsoglio, Claude Pichard, Christophe Graf, François R. Herrmann

**Affiliations:** 1Clinical Nutrition, Geneva University Hospitals and University of Geneva, 1205 Geneva, Switzerland; Julie.Mareschal@hcuge.ch (J.M.); Laurie.Karsegard@hcuge.ch (V.L.K.); Najate.Achamrah@chu-rouen.fr (N.A.); Marta.DELSOGLIO@hcuge.ch (M.D.); Claude.Pichard@unige.ch (C.P.); 2Rehabilitation and Palliative Care, Geneva University Hospitals and University of Geneva, 1205 Geneva, Switzerland; Cristophe.Graf@hcuge.ch; 3Rehabilitation and Geriatrics, Geneva University Hospitals and University of Geneva, 1205 Geneva, Switzerland; Francois.Herrmann@hcuge.ch

**Keywords:** bioelectrical impedance analysis, fat mass, fat-free mass, running performance

## Abstract

A low fat mass is associated with a good running performance. This study explores whether modifications in body composition predicted changes in running speed. We included people who underwent several measurements of body composition by bioelectrical impedance analysis between 1999 and 2016, at the “Course de l’Escalade”, taking place yearly in Geneva. Body composition was reported as a fat-free mass index (FFMI) and fat mass index (FMI). Running distances (men: 7.2 km; women: 4.8 km) and running times were used to calculate speed in km/h. We performed multivariate linear mixed regression models to determine whether modifications of body mass index, FFMI, FMI or the combination of FFMI and FMI predicted changes in running speed. The study population included 377 women (1419 observations) and 509 men (2161 observations). Changes in running speed were best predicted by the combination of FFMI and FMI. Running speed improved with a reduction of FMI in both sexes (women: ß −0.31; 95% CI −0.35 to −0.27, *p* < 0.001. men: ß −0.43; 95% CI −0.48 to −0.39, *p* < 0.001) and a reduction of FFMI in men (ß −0.20; 95% CI −0.26 to −0.15, *p* < 0.001). Adjusted for body composition, the decline in running performance occurred from 50 years onward, but appeared earlier with a body mass, FFMI or FMI above the median value at baseline. Changes of running speed are determined mostly by changes in FMI. The decline in running performance occurs from 50 years onward but appears earlier in people with a high body mass index, FFMI or FMI at baseline.

## 1. Introduction

Recreational runs are growing in popularity. In 2017, the three major popular runs in Switzerland, i.e., the “Course de l’Escalade”, “Grand Prix of Berne” and “20 km of Lausanne” brought together over 51,000, 33,000 and 27,000 participants, respectively. This corresponds to an increase of over 100% participants within 10 years and highlights the mass participation of road running among the general population [1]. The registration to recreational runs is likely used by many participants as a motivational tool for regular endurance training in order to improve health-related physical fitness. A cross-sectional study showed that the participants to the “Course de l’Escalade” aged over 50 years reported a better health status than the general population [2]. Similarly, long distance runners report a better quality of life than sedentary people or bodybuilders [3] who participate predominantly in resistance training [4].

One of the health benefits of endurance training, whether as high-intensity interval or moderate-intensity continuous training, is body weight control and reduction of fat mass [5,6]. Several cross-sectional field studies have shown the link between low fat mass and good running performance [7,8,9]. For a race distance ranging from 5 to 20 km, we have reported that this negative association occurs independently of fat-free mass, and showed that body composition was a better predictor of running performance than body mass index [10]. Longitudinal studies have confirmed the benefit of endurance training for several weeks on body composition and running performance [11,12].

However, running performance declines with ageing among elite runners, even when they maintain a high level of training [13]. Cerutti et al. showed that the running time increased by +0.11 min/year over the racing distance of 7.25 km at the Course de l’Escalade (1). This decline in performance is partly caused by a decrease in maximal aerobic capacity [13], running economy [14] and changes in running kinetics [15]. Aging is also associated with a higher body and fat mass, and a lower fat-free mass [16]. The question remains open whether body composition changes, related to ageing or potentially other factors, also affect the evolution of running performance over the years.

This longitudinal cohort study over several years aims to evaluate (1) the impact of body composition changes on the evolution of running speed in recreational runners participating in the “Course de l’Escalade”, (2) the effect of ageing on this relationship and (3) the influence of baseline body composition on the evolution of running speed. We hypothesized that the running speed decreased as body mass index and fat mass increased, that this relationship was independent of age, and that the baseline body composition had no influence on the relationship between aging and performance.

## 2. Subjects and Methods

This prospective cohort study included people > 16 years who ran in a timed city run called the “Course de l’Escalade” between 1999 until 2016, and who underwent several measurements of body composition over the years at this event. The study was accepted by the Ethical Committee of Geneva and was registered under ClinicalTrials.gov, NCT03400761.

### 2.1. City Run: the “Course de l’Escalade”

This race takes place every year during the first Saturday of December, in Geneva, Switzerland. It encompasses three major racing distances: 4.8 km for men aged 16 to 18 years and non-elite women, 7.2 km for elite men, non-elite men and elite women, and, all five years, an additional race over 18 km for both sexes. People aged 20 to 39 years had to register in the elite category if they have run at a speed of at least 12 km/h for women or 15 km/h for men in an official timed run over the previous two years. Most participants were running in the non-elite categories. Running times and running distances were provided by “Datasport AG” (Gerlafingen, CH), upon request of the Organizing Committee.

Running speed may depend on meteorological conditions. Thus, we retrieved from the internet the air temperature (°Celsius) and relative humidity (%) 2 m above ground of Geneva, from 1999 to 2016, per day and per hour [17]. Since races were performed at varying times throughout the day, we kept the mean values of each racing day, from 8 a.m. to 12 p.m.

### 2.2. Body Composition: Bioelectrical Impedance Analysis (BIA)

Since 1991, on the Saturday of the race and the preceding Friday, runners and non runners can undergo free measurements of body composition by BIA. These measurements are performed by trained staff of the nutrition unit of the Geneva University Hospitals, at a stand hold at the “Course de l’Escalade”.

First, a staff member checks with the volunteers that the body composition is measured before their run, in order to avoid invalid BIA measurements due to fluid shifts. Then, the volunteers fill in a questionnaire which includes their date of birth and habitual physical exercise. Their height and weight in light clothes are measured with a height gauge and an electronic scale, both calibrated yearly by the Geneva University Hospitals. To undergo a tetrapolar BIA measurement, the volunteers lie on their back on a medical folding bed, with arms and legs in abduction [18]. A trained staff member places four electrodes on the dorsal surface of the right wrist, hand, ankle and foot, as described elsewhere [18], applies an alternative electrical current (800 mA; 50kHz) through the BIA device, and reports the provided resistance and reactance. Body mass index is calculated as weight (kg)/height (m^2^). Fat-free mass is calculated by the Geneva formula [19], developed and validated against dual-energy X-ray absorptiometry in the population living in the Geneva area. Fat mass [kg] corresponds to the subtraction of fat-free mass from body weight. Fat-free and fat masses are divided by height squared (m^2^) to obtain fat-free mass index (FFMI) and fat mass index (FMI). The body composition results are printed, handed to each volunteer, and interpreted with a senior member of the nutrition unit.

We have been using several BIA devices over the years (BIO-Z^®^, (Spengler, Paris, France), Eugedia^®^ (Eugédia-Spengler, Cachan, France) and Nutriguard^®^ (Data Input GmbH, Darmstadt, Germany)). All devices were calibrated with the same calibration jig (CJ 4000; Xitron Technologies, City, US State abbrev. if applicable, Country), with a limit of interdevice tolerance of ±2° for phase angle and ±5 Ω for impedance, respectively. The agreement between devices for fat-free mass was 0.03 kg (95% CI: −1.7 to 2.1 kg) [10].

### 2.3. Data Merging

The running times and running distances of each person who underwent a measurement of body composition between 1999 and 2016 were used to calculate running speed in km/h. Running speed likely differs according to the distance a person has to run. In order to explore the evolution of running speed over the years, this study focused on the speed of people running in non-elite categories (7.2 km in men and 4.8 km in women). We excluded the data collected in the other running categories, as well as people who had a unique measurement of body composition over the years (Appendix A).

### 2.4. Statistics

The baseline characteristics of the included runners are presented as means ± standard deviations (SD) for continuous data, and absolute numbers and percentages for categorical data. Comparisons of continuous data between women and men were performed with unpaired *t*-tests, and comparisons of categorical data with Mann–Whitney u-tests. Speed was plotted against all measurements of body mass index, fat-free mass index or fat-mass index with superposed 5-knot splines, a non-parametric technique which smooths the trends.

For the purpose of inclusion in linear mixed regression models, participants were categorized according to their age (16–24, 25–34, 35–44, 45–54, 55–64 and >65 yearrs) and year of first measurement (1999–2001, 2002–2004, 2005–2007, 2008–2010, 2011–2013, and 2014–2016). These categorizations were performed to highlight whether there was a decrease in running performance at a specific age or year of measurement. Univariate linear mixed regressions were calculated between speed and body mass index, FFMI or FMI. We then performed five different multivariate linear mixed regression models for each sex. They examined the impact of body mass index, FFMI, FMI or the combination of FFMI and FMI on the evolution of running speed and were adjusted for categories of age and year of measurement, air temperature and relative humidity. We compared the abilities of these models to predict the evolution of speed by calculating the Bayesian information criterion (BIC). The lower the BIC, the better is the predictive ability of the model. The BIC was also compared between nested models with a likelihood ratio test. Using the best fitting model, we calculated the predicted speed for each participant.

In order to evaluate whether running performance dropped at a specific age, predicted speed was plotted against age, with superposed 5-knot splines. To determine whether baseline body composition had an impact on the evolution of running speed, we split the participants into two groups according to the median values of the body mass index, FFMI or FMI at baseline. We repeated the plotting of predicted speed vs. age using these categorizations.

## 3. Results

The study population included 377 women (1419 observations, minimum 2, maximum 15, mean 3.8 by participant) and 509 men (2161 observations, minimum 2, maximum 16, mean 4.2 by participant), whose baseline characteristics are shown in Table 1. Men and women differed for all anthropometric characteristics besides age. Figure 1 shows the running speed plotted against all measurements of body mass index, FFMI and FMI, with superposed 5-knot splines. It highlights a better running speed as body mass index, and especially FMI, decrease.

The results of the univariate linear mixed regressions are shown on Appendix A. Based on the Baysian information criterion (BIC), they highlight that running speed is slightly better predicted by changes in FMI and FFMI than body mass index in women (BIC = 3458 vs. 3501, respectively) and men (BIC = 5755 vs. 5794, respectively). The multivariate linear mixed regression models demonstrate that running speed increased as body mass index or FMI decreased in women (Table 2) and men (Table 3). Interestingly, running speed improved with a reduction of FFMI in men but not in women. Whatever model considered, the more recent the year of measurement, the lower the running speed. Although all models predicted running speed (*p* < 0.001 for all models), the best predictive model was the one that included simultaneously FFMI and FMI (model 5), as it showed the lowest Bayesian information criterion and significantly differed from the other models (Appendix A).

There was no linear decrease of running speed with age categories and years of measurement in any multivariate linear mixed regression model. Figure 2 shows a decline in running performance since the age of 50 years in both genders but especially in men. Male and female runners with a low body mass index, FFMI and FMI at baseline experienced a stability of their performance until 50 years and a decrease thereafter (Appendix A). However, male and female runners with a body mass index, FFMI and FMI above the median values showed a decrease of their performance already from 30 years onward. This decrease was statistically significant only in women for body mass index and FMI.

## 4. Discussion

This longitudinal cohort study showed that running speed over 5 km in women and 7 km in men increased as FMI, and, to a lesser extent, body mass index decreased. The impact of FFMI was gender-specific, since running speed improved with a decline in FFMI in men but was not influenced by changes in FFMI in women. When adjusting for body composition, the decline in running performance occurred from 50 years onward. Participants with a body mass index, FFMI or FMI above the median value at baseline showed an earlier decline of their performance with ageing than those with values under the median.

Cross-sectional studies have shown that running performance is inversely associated with fat mass [7,8,9,10]. Ravnholt et al. reported that untrained men and women who undertook a seven-week high-intensity interval training three times per week decreased their fat mass and increased their fat-free mass, and subsequently improved their running performance over 1.5 and 3 km [11]. Similarly, another study showed that a six-week sprint interval training of 30 s or an endurance training of 30–60 min thrice weekly, decreased fat mass and increased fat-free mass [12]. These changes were associated with an improved running performance over 2 km in men and women [12]. Our study confirms that an improvement of running speed in recreational runners is determined mostly by a decrease in FMI. However, in contrast to the afore-mentioned studies, evolution of running speed over 5 or 7 km was negatively affected by the FFMI in men. The reason for this finding is unclear. One hypothesis is that a high fat-free mass may hamper running performance in distances over 3 km, as it requires higher weight bearing. As men have a higher fat-free mass than women, we could hypothesize that they can lose more fat-free mass to improve their running speed, without compromising their health. In line with this hypothesis, men were reported to have a higher anaerobic power reserve than women [20]. This result suggests that men have a muscle mass which is more efficient metabolically than women, and that they need less muscle mass than women for a given effort.

Cerutti et al. showed a peak performance at the “Course de l’Escalade” around the age of 20 years, then a slow decrease until 50 years and a more rapid reduction thereafter, but they did not adjust their results for body composition [1]. Our study corroborates the decline in running performance from 50 years onward, when adjusted for body composition and suggests that elements other than body composition contribute to the decrease of running performance. For instance, VO_2max_ was reported to decrease linearly with age, by 5 to 7% per decade, in elite distance runners [13]. Running economy, defined as the oxygen consumption for a given running velocity, also declined in elite distance runners followed over 45 years [14]. Finally, Korhonen et al. showed that stride length, ground reaction force at push-off phase and dynamic strength of leg extensors decreased around age 40 to 50 years, which slowed running speed [15]. Thus, decreased cardio-respiratory fitness, as well as alterations of running kinetics, may explain the decreased running performance after 50 years.

Participants with a high body mass index, FFMI or FMI at baseline showed an earlier decline of their performance with ageing than those with lower values. A high body mass index and FMI may reflect a more sedentary lifestyle of these participants. It thus leads to the hypothesis that a sedentary lifestyle results in an earlier decline in running speed as compared to a physically active lifestyle. As a corollary, participants with a low body mass index, FFMI and FMI, which may reflect endurance training, could maintain their running speed longer over the years than their sedentary counterparts. Thus, our study confirmed a previous study reporting that body composition changed less with aging in track athletes than in healthy sedentary people [21].

The strength of the study relies on its longitudinal design, the high number of participants, and the precise running speed. The main limitations are the absence of information on other elements which may influence body composition changes, such as physical activity, energy intakes, co-morbidities or pharmaceutical drugs and the lack of information on physiological parameters as VO_2max_. However, our subjects are apparently healthy and changes in energy balance should translate into changes of body composition. The body composition was measured by several BIA devices whose agreement may be questioned. However, all devices had been cross-calibrated with the same calibration jig and we used a single formula, developed in the Geneva population and validated against dual-energy X-ray absorptiometry, to derive fat-free mass [19]. We also acknowledge a selection bias, as only volunteers were measured, but this bias should not influence the relationship between changes in body composition and running speed in recreational runners.

## 5. Conclusions

A decrease of FMI in both sexes and a decrease of FFMI in men predict an improvement of running speed over 5 to 7 km. The decline of running performance with aging occurs from 50 years onward in both sexes, but appears earlier in people with a high body mass index, FFMI or FMI at baseline. This study highlights the importance of following body composition in ambitious recreational runners who want to maintain their performance as long as possible. The results of this study and regular monitoring of body composition could encourage the long-term participation of the general population in endurance activities and timed city runs in order to maintain body composition, not only for performance, but also for health benefits.

## Figures and Tables

**Figure 1 nutrients-11-00701-f001:**
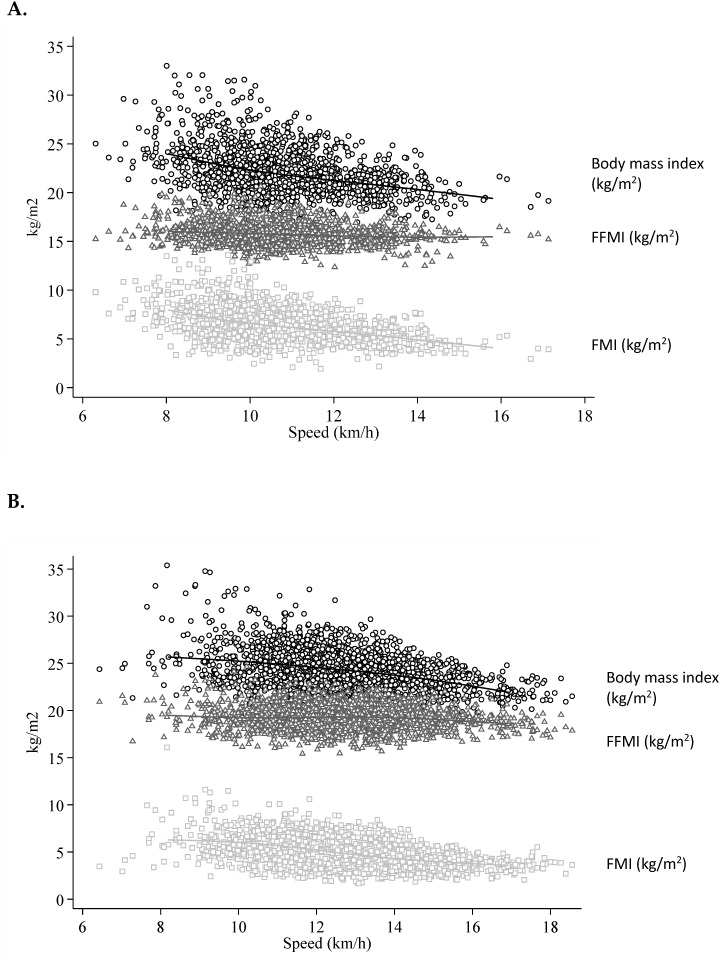
Two-way scatterplot showing the actual running speed vs. all measurements of body mass index (**○**), fat-free mass index (**Δ**) and fat mass index (**□**) in women (**A**; 377 women, 1419 observations) and men (**B**; 509 men, 2161 observations].

**Figure 2 nutrients-11-00701-f002:**
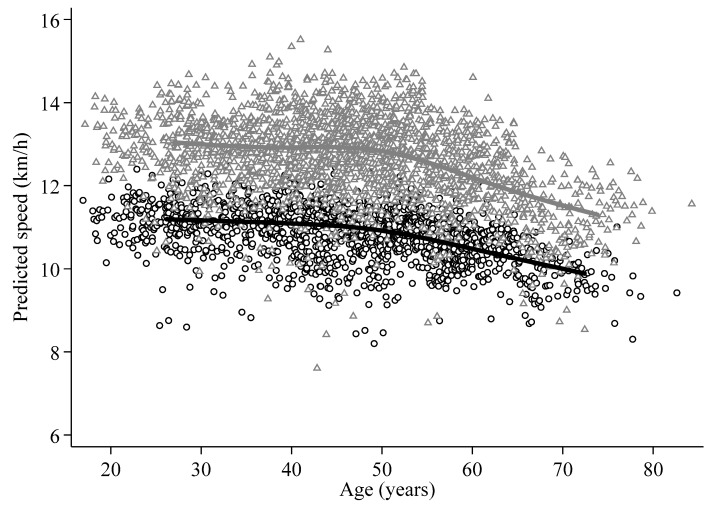
Two-way scatterplot showing the predicted running speed of each participant according to model 5 of multivariate linear mixed regression models against age, in women (○; 377 women, 1419 observations) and men (Δ; 509 men, 2161 observations).

**Table 1 nutrients-11-00701-t001:** Characteristics of the study population at baseline (*n* = 886).

	All					Women					Men					
Variables	*n*	%	Mean		SD	*n*	%	Mean		SD	*n*	%	Mean		SD	*p* *
**Continuous**																
Number of BIA measurements	886	100	4.5	±	3.0	377	43	4.0	±	2.7	509	57	4.8	±	3.1	<0.001
Age at measurement [years]	886	100	41.7	±	12.0	377	43	41.1	±	12.2	509	57	42.0	±	11.9	0.265
Body weight [kg]	886	100	68.2	±	11.2	377	43	59.2	±	8.0	509	57	74.8	±	8.2	<0.001
Height [cm]	886	100	170.9	±	9.5	377	43	163.4	±	6.7	509	57	176.4	±	7.1	<0.001
Body mass index [kg/m^2^]	886	100	23.2	±	2.5	377	43	22.1	±	2.5	509	57	24.0	±	2.1	<0.001
Fat-free mass index [kg/m^2^]	886	100	17.7	±	2.1	377	43	15.7	±	1.2	509	57	19.2	±	1.3	<0.001
Fat mass index [kg/m^2^]	886	100	5.6	±	1.7	377	43	6.4	±	1.7	509	57	4.9	±	1.4	<0.001
Fat mass [%]	886	100	23.7	±	6.3	377	43	28.7	±	4.9	509	57	20.1	±	4.5	<0.001
Running speed [km/h]	886	100	12.0	±	1.9	377	43	10.9	±	1.6	509	57	12.9	±	1.7	<0.001
**Categorical**																
Age [years]																0.262
16–24	75	8				39	10				36	7				
25–34	197	22				84	222				113	22				
35–44	280	32				116	31				164	32				
45–54	212	24				90	24				122	24				
55–64	97	11				38	10				59	12				
≥65	26	3				11	3				15	3				
Body mass index [kg/m^2^]																
<18.5	11	1				10	2				1	1				<0.001
18.5–24.9	685	77				326	87				357	70				
25–29.9	184	21				38	10				146	28				
> or = 30	8	1				3	1				5	1				
Year of measurement [years]																0.042
1999–2001	283	32				111	30				171	34				
2002–2004	191	22				76	20				115	22				
2005–2007	131	15				56	15				75	15				
2008–2010	128	14				58	15				70	14				
2011–2013	81	9				43	11				38	7				
2014–2016	73	8				33	9				40	8				

* unpaired *t*-test or Mann–Whitney u-test; BIA: bioelectrical impedance analysis.

**Table 2 nutrients-11-00701-t002:** Multivariate mixed linear regression to predict changes in running speed in women (377 participants, 1419 observations).

	Model 1				Model 2				Model 3				Model 4				Model 5			
	ß	95% CI		*p*	ß	95% CI		*p*	ß	95% CI		*p*	ß	95% CI		*p*	ß	95% CI		*p*
Age [years]																				
15–24	0.00				0.00				0.00				0.00				0.00			
25–34	−0.07	−0.34	0.20	0.615	−0.02	−0.28	0.25	0.904	−0.07	−0.34	0.21	0.636	−0.03	−0.29	0.23	0.814	−0.03	−0.29	0.23	0.821
35–44	−0.06	−0.38	0.25	0.687	−0.05	−0.35	0.24	0.720	−0.06	−0.37	0.25	0.708	−0.11	−0.40	0.18	0.460	−0.11	−0.40	0.18	0.469
45–54	−0.20	−0.54	0.13	0.232	−0.17	−0.49	0.14	0.279	−0.19	−0.53	0.14	0.253	−0.25	−0.56	0.06	0.108	−0.25	−0.56	0.06	0.114
55–64	−0.51	−0.88	−0.14	0.007	−0.51	−0.86	−0.16	0.004	−0.51	−0.88	−0.14	0.007	−0.58	−0.93	−0.24	0.001	−0.58	−0.92	−0.24	0.001
≥65	−1.08	−1.52	−0.63	<0.001	−1.11	−1.53	−0.69	<0.001	−1.08	−1.53	−0.64	<0.001	−1.17	−1.58	−0.76	<0.001	−1.16	−1.57	−0.75	<0.001
Body mass index [kg/m^2^]					−0.21	−0.24	−0.17	<0.001												
Fat−free mass index [kg/m^2^]									−0.05	−0.12	0.02	0.163					−0.01	0.80	−0.07	0.056
Fat mass index [kg/m^2^]													−0.31	−0.35	−0.27	<0.001	−0.31	−0.35	−0.27	<0.001
Year of measurement [years]																				
1999–2001	0.00				0.00				0.00				0.00				0.00			
2002–2004	0.01	−0.13	0.16	0.858	0.02	−0.12	0.16	0.811	0.03	−0.12	0.18	0.687	−0.09	−0.23	0.05	0.216	−0.08	−0.23	0.06	0.239
2005–2007	−0.11	−0.26	0.04	0.163	−0.03	−0.17	0.12	0.740	−0.09	−0.24	0.07	0.273	−0.12	−0.26	0.03	0.114	−0.11	−0.26	0.03	0.132
2008–2010	−0.16	−0.41	0.08	0.194	−0.03	−0.27	0.21	0.793	−0.13	−0.38	0.12	0.320	−0.19	−0.42	0.05	0.117	−0.18	−0.42	0.06	0.136
2011–2013	−0.57	−0.81	−0.33	<0.001	−0.43	−0.66	−0.19	<0.001	−0.54	−0.78	−0.29	<0.001	−0.55	−0.78	−0.32	<0.001	−0.55	−0.78	−0.31	<0.001
2014–2016	−0.54	−0.76	−0.31	<0.001	−0.38	−0.60	−0.16	0.001	−0.52	−0.74	−0.29	<0.001	−0.44	−0.65	−0.23	<0.001	−0.43	−0.65	−0.22	<0.001
Temperature [°C]	−0.02	−0.03	0.00	0.027	−0.02	−0.03	0.00	0.037	−0.02	−0.03	0.00	0.045	−0.02	−0.04	−0.01	0.001	−0.02	−0.04	−0.01	0.001
Relative humidity [%]	0.00	0.00	0.01	0.192	0.01	0.00	0.01	0.057	0.00	0.00	0.01	0.151	0.00	0.00	0.01	0.194	0.00	0.00	0.01	0.188

CI: confidence interval

**Table 3 nutrients-11-00701-t003:** Multivariate mixed linear regression to predict changes in running speed in men [509 participants, 2161 observations].

	Model 1				Model 2				Model 3				Model 4				Model 5			
	ß	95% CI		*p*	ß	95% CI		*p*	ß	95% CI		*p*	ß	95% CI		*p*	ß	95% CI		*p*
Age [years]																				
15–24	0.00				0.00				0.00				0.00				0.00			
25–34	0.12	−0.15	0.39	0.369	0.24	0.00	0.49	0.054	0.13	−0.14	0.40	0.337	0.24	−0.01	0.49	0.064	0.26	0.01	0.50	0.042
35–44	0.27	−0.03	0.58	0.079	0.47	0.19	0.75	0.001	0.29	−0.01	0.59	0.061	0.44	0.16	0.72	0.002	0.48	0.20	0.76	0.001
45–54	0.25	−0.08	0.58	0.144	0.50	0.19	0.81	0.001	0.29	−0.04	0.62	0.090	0.40	0.10	0.71	0.010	0.48	0.18	0.79	0.002
55–64	−0.36	−0.73	0.01	0.056	−0.13	−0.47	0.21	0.453	−0.34	−0.70	0.03	0.073	−0.22	−0.56	0.12	0.211	−0.14	−0.48	0.19	0.405
≥65	−1.28	−1.72	−0.84	<0.001	−1.03	−1.44	−0.63	<0.001	−1.27	−1.70	−0.83	<0.001	−1.11	−1.52	−0.71	<0.001	−1.04	−1.44	−0.64	<0.001
Body mass index [kg/m^2^]					−0.35	−0.38	−0.31	<0.001												
Fat-free mass index [kg/m^2^]									−0.16	−0.22	−0.10	<0.001					−0.20	−0.26	−0.15	<0.001
Fat mass index [kg/m^2^]													−0.42	−0.47	−0.38	<0.001	−0.43	−0.48	−0.39	<0.001
Year of measurement [years]																				
1999–2001	0.00				0.00				0.00				0.00				0.00			
2002–2004	−0.25	−0.39	−0.11	<0.001	−0.25	−0.38	−0.12	<0.001	−0.19	−0.33	−0.05	0.006	−0.39	−0.52	−0.26	<0.001	−0.32	−0.45	−0.19	<0.001
2005–2007	−0.37	−0.52	−0.22	<0.001	−0.23	−0.37	−0.09	0.001	−0.30	−0.45	−0.15	<0.001	−0.38	−0.52	−0.24	<0.001	−0.30	−0.43	−0.16	<0.001
2008–2010	−0.66	−0.89	−0.42	<0.001	−0.45	−0.66	−0.23	<0.001	−0.53	−0.77	−0.29	<0.001	−0.72	−0.94	−0.50	<0.001	−0.57	−0.79	−0.35	<0.001
2011–2013	−1.21	−1.44	−0.97	<0.001	−0.91	−1.12	−0.69	<0.001	−1.08	−1.31	−0.84	<0.001	−1.16	−1.38	−0.94	<0.001	−1.01	−1.23	−0.79	<0.001
2014–2016	−1.19	−1.40	−0.97	<0.001	−0.96	−1.16	−0.76	<0.001	−1.11	−1.33	−0.89	<0.001	−1.09	−1.29	−0.89	<0.001	−1.00	−1.20	−0.80	<0.001
Temperature [°C]	−0.03	−0.04	−0.01	0.001	−0.03	−0.04	−0.01	<0.001	−2.68	−0.04	−0.01	0.007	−0.04	−0.05	−0.02	<0.001	−0.03	−0.05	−0.02	<0.001
Relative humidity [%]	0.00	0.00	0.01	0.218	0.01	0.00	0.01	0.014	1.97	0.00	0.01	0.048	0.00	0.00	0.01	0.559	0.00	0.00	0.01	0.115

CI: confidence interval

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
