# Peer review of "An Increase in Fat Mass Index Predicts a Deterioration of Running Speed"

_nutrients, 2019, doi:10.3390/nu11030701_

Reviewer 1 Report

In this longitudinal study, authors tried to explore the association between body composition and running speed. Authors found that improved running speed was associated with decreased FMI in men and women, and decreased FFMI in men, not in women. The observation through the statistics are interesting, however no information about the nutritional interventions were described in this study. Several other concerns also should be addressed for further consideration of publication.

Major:

1.       Despite the interesting findings on association between running speed and body composition, this study didn’t provide enough information about any possible factors that are involved in changes of either running speed or body composition.

Others:

1.       Please revise the title to be specific to the purpose/conclusion of the study. Avoid superfluous words in the Title.

2.       Page 1, Line 15, what does it mean ‘all people?’ Please provide exact number or rephrase the sentence. Line 16, ‘occurring’ replace with other suitable word.

3.       Page 1, Line 28, ‘dictate’ sounds very strong. As this study didn’t provide direct evidence, please replace with other suitable word. 

4.       Page 2, Line 26. Please check, what is all people > 16 years? 

5.       Page 2, Lines 41-42. This sentence means authors included meteorological data from 8 am to 12 pm. It is not clear why data kept for a particular time, when event took longer time.

6.       In this study, what does body composition measurements means…only BMI, FMI and FFMI or any other variables? Does any coauthors involved in measurements of body composition?

7.       Were the protocols and instruments were the same from 1999 to 2016. If different devises used for the same measurements, please explain the protocol specific to the each device that used in this study.

8.       I wonder, is there any coauthor continuously participating in this study from 1999 to 2016?

9.       Please check the age group categories as described in Page 3 (16-24) and Page 5 (Table 1, 15-24).

10.   The Results part could be strengthen to emphasize the importance of the data. The description of Figure 1 is very simple. Please explain that does the figure/data mean.

11.   Figure 1. The labeling of lines (BMI, FMI, FFMI) could mark in figure itself rather than legend.  

12.    Figure 2. Scatter plot showing the running speed of men and women with identical symbols (round circles). From this figure it is hard to identify the data of men or women, please change the symbols.  

13.   The discussion part could be strengthening according to the findings.  

14.   It is hard to collect the nutrition data for such a large scale population in this longitudinal study. Despite, how authors could convince their findings with possible reasons for increasing or decreasing the running speed over a period of time. To be specific, how these findings can attract the readers of “Nutrients” journal?

Author Response

Answers to reviewer 1

In this longitudinal study, authors tried to explore the association between body composition and running speed. Authors found that improved running speed was associated with decreased FMI in men and women, and decreased FFMI in men, not in women. The observation through the statistics are interesting, however no information about the nutritional interventions were described in this study. Several other concerns also should be addressed for further consideration of publication.

Thank you very much for these encouraging comments. This cohort study is indeed observational and there is no nutritional intervention.

1.   Despite the interesting findings on association between running speed and body composition, this study didn’t provide enough information about any possible factors that are involved in changes of either running speed or body composition.

 Indeed, we have not other elements than running speed or body composition, although some may have been interesting. Nevertheless, we would like to point out that this was a field study performed yearly on the 2 racing days, on a stand at the “Course de l’Escalade”. In our setting, it is not possible to measure physiological factors such as cardiorespiratory fitness (very time-consuming, requiring generally sophisticated devices and much manpower). We could have collected data on nutritional intakes or physical activity through questionnaires, but their validity would have been uncertain, because reported, and would have had to be checked individually by the staff with each participant, which is again very time-consuming. We agree that between 1999 and 2016, many factors may have changed, but some of them reflect at least in body composition changes (as physical activity, calorie balance). ), and we corrected for the period effect in multiple regression models.

2.   Please revise the title to be specific to the purpose/conclusion of the study. Avoid superfluous words in the Title.

 The introduction states that the article “ aims at evaluating 1) the impact of body composition changes on the evolution of running speed in recreational runners participating to the “Course de l’Escalade” (page 4, line 3-5). The first sentence of the conclusion is: “The evolution of running speed over 5 to 7 km is dictated mostly by the changes of fat mass index”.

 As requested by the reviewer and according also to the suggestions of reviewer 2 (question 1), we have shortened and changed the title: “A decrease in fat mass index predicts an improvement of running speed”.

3.      Page 1, Line 15, what does it mean ‘all people?’ Please provide exact number or rephrase the sentence. Line 16, ‘occurring’ replace with other suitable word.

 “All people” means that we considered all people who had several body composition measurements during the “Course de l’Escalade” at our stand, and run 4.8 km (women) or 7.2 km (men). As mentioned in the chapter data merging (page 3, line 26-28), “we excluded the data collected in the other running categories, as well as people who had a unique measurement of body composition over the years”.  However, the exclusion criteria did not fit in the abstract because it is limited to 200 words. The details of inclusion and exclusion criteria are shown on supplemental figure 1.

 In response to the suggestion of the reviewer, we changed the sentence, page 1, line 15 to: “We included people who underwent several measurements of body composition…”

“Occurring” on line 16 was changed to “taking place”.

 4.   Page 1, Line 28, ‘dictate’ sounds very strong. As this study didn’t provide direct evidence, please replace with other suitable word.

 As requested, we changed the word “dictated” by “determined”.

 5.   Page 2, Line 26. Please check, what is all people > 16 years?

All people > 16 years means that we included, as mentioned, every person over 16 years at the time of his first body composition measurement and who run the course de l’Escalade that same year. As we evaluate changes of body composition and running speed, every person has to have at least another measurement of body composition at a following edition of the “Course de l’Escalade” with a simultaneous running time.

 To make the sentence clearer, we changed it to: “This prospective cohort study included people > 16 years…”

 6.   Page 2, Lines 41-42. This sentence means authors included meteorological data from 8 am to 12 pm. It is not clear why data kept for a particular time, when event took longer time.

 Indeed we kept the mean meteorological data from 8 am to 12 pm of the racing day of the participant, which matches the time of the races. No race was performed during the night, so there was no use of considering the meteorological data during the night.

 7.   In this study, what does body composition measurements means…only BMI, FMI and FFMI or any other variables? Does any coauthors involved in measurements of body composition?

 In this study, body composition refers to FMI and FFMI, as measured by BIA. All co-authors were involved at least one year in the measurements of body composition at the “Course de l’Escalade”. L. Genton, V. L. Karsegard and C. Pichard were present at all editions since 1999 and responsible for the training of the participating staff.

8.    Were the protocols and instruments were the same from 1999 to 2016. If different devises used for the same measurements, please explain the protocol specific to the each device that used in this study.

The protocols of BIA measurement (position of the person, placement and type of electrodes) were the same from 1999 to 2016, whatever device was used. They are the same as those used for patients in the Geneva University Hospitals (C. Graf. V. L. Karsegard, A. Spoerri, A.-M. Makhlouf, S. Ho, F. R. Herrmann, L. Genton. Am J Clin Nutr, 2015, 101(4):760-7).

As mentioned on page 3, line 16-20, we used three different tetrapolar BIA devices over the years during the “Course de l’Escalade”. The electrical parameters (impedance, phase angle) were all cross-calibrated with the same calibration jig. The electrical parameters provided by the BIA devices were used to calculate the fat mass and fat-free mass by a single formula, validated in the population of the Geneva area. Thus, we did not rely on device-specific formulas, which are often implemented automatically in BIA devices.

9.    I wonder, is there any coauthor continuously participating in this study from 1999 to 2016?

As answered in question 7, all co-authors were involved at least one year in the measurements of body composition at the “Course de l’Escalade”. L. Genton, V. L. Karsegard and C. Pichard were present at all editions since 1999 and responsible for the training of the participating staff. The measurements will still go on over the next years, as the Nutrition Unit is supported financially and logistically by the Geneva University Hospital for this event.

 10. Please check the age group categories as described in Page 3 (16-24) and Page 5 (Table 1, 15-24).

 We thank the reviewer for his careful reading. The age group category in page 5 was a mistake and was changed to “16-24” as in page 3.

 11. The Results part could be strengthen to emphasize the importance of the data. The description of Figure 1 is very simple. Please explain that does the figure/data mean.

 The results section aims to describe objectively our results. The importance of the data is emphasized in the discussion.

 Figure 1 shows that the lower the body mass index, the higher is the running speed in both sexes. When considering the components of body mass index, it highlights that especially a low FMI determines a better running speed.

To improve the description of the figure, we added the following sentence on page 4, line 11: “It highlights a better running speed as body mass index, and especially FMI, decrease.”

12.Figure 1. The labeling of lines (BMI, FMI, FFMI) could mark in figure itself rather than legend. 

 As suggested by the reviewer, the labeling was additionally marked on the figure itself.

 13.Figure 2. Scatter plot showing the running speed of men and women with identical symbols (round circles). From this figure it is hard to identify the data of men or women, please change the symbols

 As suggested by the reviewer, the symbols for men were changed to triangles, which are different from the circles used for women.

 14. The discussion part could be strengthening according to the findings

Thank you for this comment. We have already summarized the results in the first paragraph of our discussion and thus highlighted the important points. In the next paragraphs, we have commented on our findings, according to the literature. In the conclusion, we have again stressed the main findings. There are only few longitudinal studies (over several years) on body composition and physical activity/running and thus we do not see how we should strengthen the discussion more according to the findings.

 15.It is hard to collect the nutrition data for such a large scale population in this longitudinal study. Despite, how authors could convince their findings with possible reasons for increasing or decreasing the running speed over a period of time. To be specific, how these findings can attract the readers of “Nutrients” journal?

 Nutrition and nutrients, as well as physical activity, impact on energy balance and thus on body weight and body composition. Body composition is considered as a marker of the nutritional state, as demonstrated by its inclusion in the diagnostic criteria of malnutrition (Cerderholm et al, Clin Nutr 2019, 38(1): 1-9). “Nutrients” has already published numerous articles on body composition, showing the interest of the journal and his readers for this topic.

As a low muscle mass and a high fat mass are associated with negative outcomes, lifestyle changes which improve body composition, are of particular interest. Among them, physical activity plays an essential role as shown for instance in the position stand of the American College of Sports Medicine (Chodzko-Zjako et al, Med Sci Sports Exerc 2009, 41: 1510-30). In our opinion, one way to stay motivated in physical activity is to participate to popular sport events, like city runs with the aim of improving health-related fitness (for instance body composition) or performance.

Thus, if the readers of “Nutrients” are interested by physical fitness and its component body composition as part of their overall health and which integrates energy balance, we think that they should be interested in that topic.

Of note, in the general population, the link between body composition and running speed attracts much interest, as shown in the many BIA measurements we are performing every year within 2 days at the “Course de l’Escalade”.

Reviewer 2 Report

General comments

This was a well written manuscript, with a clear purpose – the findings are novel, and pertinent to a growing population of recreational runners. I have some general comments listed here, and a more thorough breakdown of each section below.

Please can you confirm and ensure consistency throughout for the abbreviation for fat free mass index, as this is listed as FFMI: Fat free mass at the start of the article, but used as fat free mass index throughout the discussion. It is excellent to see trial registration for this study; this is very encouraging for an exercise science study. Please remove page 10, if possible. Please be aware that there are inconsistencies in how V̇O2max is written in the text. My understanding is that the correct method is to capitalise VO with an operator dot above the V, and subscript 2max. The word evolution is used consistently throughout and on occasion interchangeably with decline. I would advise an alternative word that better supports the direction of the trend in running performance seen as one ages e.g. decline, deterioration or downward progression.

 introduction

Page 1 Line 36: Please consider changing the word ‘democratization’ to ‘age group participation’ or ‘mass participation’, as democratization would suggest that the organisation of events by athletes running them has led to improvements in attendance/performance. This may be the case but is not supported by the literature cited.

Page 2 Line 2: Please consider adding ‘who participate predominantly in resistance training’ after bodybuilders and support with further references.

Page 2 Line 5: please state the direction of the relationship between fat mass and running performance in this sentence and change ‘good’ to ‘improved’, as good is a vague term here

Page 2 Line 14: have you any other factors that may explain a decline in performance, apart from the already listed decrease in aerobic capacity? How does running economy interact with this as one ages? There are likely other metabolic, mechanical or physiological predictors of performance that decline as one ages but can be maintained with training – see Everman et al., 2018 for a 45 year follow up study in elite distance runners.

Page 2 Line 18: please change ‘aims at’ to ‘aims to evaluate’

 Methods

Page 2 Line 33: please can you drop the term ‘all five years’. It doesn’t seem to fit or make sense in its current position within the text

Page 2 Line 36: please reconsider the tense of the phrase ‘Most participants are running in the non-elite categories.’

Page 2 Line 39: this should read ‘the internet’

Page 2 Line 41: please change ‘all over the day’ to ‘at varying times throughout the day’

Page 3 Line 20: you have a really small error between BIA devices, but the confidence limit seems problematic, especially as most race participants may lose this amount of weight as sweat losses over race durations of ~45-60min in length. I would question the reliability of the devices, at least within the population discussed.

Results

Page 4 Line 18/19: ‘the lower was the running speed’ can be amended to ‘the lower the running speed’

Discussion

‘…men increased as FMI, and to a lesser extent, body mass index.’ this is an incomplete sentence currently. Please complete this sentence.

‘submitted to a seven-week high-intensity interval training thrice weekly’ may read better as ‘men and women who undertook seven-week high-intensity interval training programme, performed three times per week, decreased…’

As above ‘three times per week’ may read better than ‘thrice’. Whilst this is grammatically correct, the term is not necessarily in common parlance and may be difficult for those whom English is not their primary language.

Please provide a reference or two to support your hypothesis that men may lose more fat free mass than women and improve their running speed. This is really interesting, and I think you’re correct in moving towards this hypothesis, it can be developed quite nicely with some supporting references.

‘Cerutti et al, who showed’ can be amended to ‘Cerutti et al, showed’ or consider the word reported here

‘could maintain longer their running speed…’ should be changed to something that re-emphasises the alternative i.e. sedentary lifestyle e.g. ‘could maintain their running speed longer over the years than their sedentary counterparts.’

‘The main limitations is’ should read ‘the main limitations are…’

Where you mention drugs, you may wish to clarify that these are pharmaceutical as opposed to recreational

Your final sentence is good, but please can you clarify the level of runners you are assessing. This will make for a stronger closing statement e.g. ‘but this bias should not influence the relationship between changes in body composition and running speed in recreational runners.’

Conclusion

You mention gender specific effects in your discussion, and your results highlight these nicely. Please refer to them in your conclusion and be specific as to which variables negatively affect running performance in which sex. This will make for a more thorough conclusion.

Consider changing ‘following body composition in ambitious runners…’ to ‘assessing body composition in recreational runners…’

Figures and tables

Supplemental figure 1: Please check that all numbers are correct – there seems to be a large discrepancy in the exclusion figures?

Supplemental figures are excellent and support your conclusions well – if possible it may be worth including them, and developing the discussion accordingly as this would bring more depth to the data presented/interpreted.

Author Response

Answers to reviewer 2

This was a well written manuscript, with a clear purpose – the findings are novel, and pertinent to a growing population of recreational runners. I have some general comments listed here, and a more thorough breakdown of each section below.

Thank you very much for your nice comments.

1.   Please can you confirm and ensure consistency throughout for the abbreviation for fat free mass index, as this is listed as FFMI: Fat free mass at the start of the article, but used as fat free mass index throughout the discussion. It is excellent to see trial registration for this study; this is very encouraging for an exercise science study. Please remove page 10, if possible. Please be aware that there are inconsistencies in how V̇O2max is written in the text. My understanding is that the correct method is to capitalise VO with an operator dot above the V, and subscript 2max. The word evolution is used consistently throughout and on occasion interchangeably with decline. I would advise an alternative word that better supports the direction of the trend in running performance seen as one ages e.g. decline, deterioration or downward progression.

The distinction between fat-free mass and FFMI was done on purpose. In the introduction, the term fat-free mass was used because the mentioned articles have not divided fat-free mass by body height (m)2 to obtain FFMI. Thus, fat-free mass was not normalized for body height.

In the discussion, when we mention FFMI or FMI, we are talking about our own results, and thus body composition is normalized for body height. However, we recognize that we have used the term fat-free mass once instead of FFMI in the discussion when talking about our results, and this was corrected (page 12, second paragraph of the discussion).

Thank you for the very nice comment regarding the trial registration.

We removed page 10 as requested.

A suggested, we harmonized the writing of the term “V̇O2max” in the manuscript, which appears twice in the discussion.

We have used the term “evolution” in the title, the abstract the introduction and conclusion.

-       We have changed the title to “A decrease in fat mass index predicts an improvement of running speed”.

-       In the introduction, we prefer to keep the term evolution as we do not know yet in what direction the running performance changes.

-       In the conclusion of the abstract and of the manuscript, we prefer to keep the term “evolution” in the first sentence, in order to avoid a repetition with the second sentence, where the term “decline” is already used.

2.   Page 1 Line 36: Please consider changing the word ‘democratization’ to ‘age group participation’ or ‘mass participation’, as democratization would suggest that the organisation of events by athletes running them has led to improvements in attendance/performance. This may be the case but is not supported by the literature cited.

As requested, the term “democratization” was changed to “mass participation”.

 3.   Page 2 Line 2: Please consider adding ‘who participate predominantly in resistance training’ after bodybuilders and support with further references.

 As suggested, we added the word “who participate predominantly in resistance training”, with a reference showing that bodybuilding consist of resistance training.  

 4.   Page 2 Line 5: please state the direction of the relationship between fat mass and running performance in this sentence and change ‘good’ to ‘improved’, as good is a vague term here

 As suggested, we changed the sentence to “A low fat mass is associated with a good running performance”.

 5.   Page 2 Line 14: have you any other factors that may explain a decline in performance, apart from the already listed decrease in aerobic capacity? How does running economy interact with this as one ages? There are likely other metabolic, mechanical or physiological predictors of performance that decline as one ages but can be maintained with training – see Everman et al., 2018 for a 45 year follow up study in elite distance runners.

 Factors as the decrease in aerobic capacity and changes in running kinetics have been detailed in the discussion. Thank you for the reference regarding the decline in running economy with aging, although the authors have not measured any running performance in the mentioned article.

In order to be more complete in the introduction, we have added on page 2, line 17-18:  “This decline in performance is partly caused by a decrease in maximal aerobic capacity [12], running economy {Everman, 2018 #795} and changes in running kinetics [17] .

In order to be consistent with the introduction, the article of Everman et al. has been detailed in the discussion: “Running economy, defined as the oxygen consumption for a given running velocity, also declined in elite distance runners followed over 45 years {Everman, 2018 #795}.”

 6.   Page 2 Line 18: please change ‘aims at’ to ‘aims to evaluate’

 The change was made as requested.

 7.   Page 2 Line 33: please can you drop the term ‘all five years’. It doesn’t seem to fit or make sense in its current position within the text

 We cannot drop this term as the additional race is occurring only all five years. If we place it in another position in the sentence, the reader may get confused and think that all races are taking place only every five years.

 Thus we prefer to keep this term at the present place.

 8.   Page 2 Line 36: please reconsider the tense of the phrase ‘Most participants are running in the non-elite categories.’

 The tense was changed to the past.

 9.   Page 2 Line 39: this should read ‘the internet’

 This was changed as requested.

 10. Page 2 Line 41: please change ‘all over the day’ to ‘at varying times throughout the day’

 This was changed as requested.

 11. Page 3 Line 20: you have a really small error between BIA devices, but the confidence limit seems problematic, especially as most race participants may lose this amount of weight as sweat losses over race durations of ~45-60min in length. I would question the reliability of the devices, at least within the population discussed.

 As mentioned in the method (page 3, line 5-6), the BIA measurements were performed before the run, thus the problem of sweat loss related to exercise is not relevant.

 The confidence interval for FFM agreement between BIA devices can be viewed as a limitation. However, it is a field method with a good inter-individual reproducibility (C. Graf. V. L. Karsegard, A. Spoerri, A.-M. Makhlouf, S. Ho, F. R. Herrmann, L. Genton. Am J Clin Nutr, 2015, 101(4):760-7 or Earthman, JPEN, 201539(7): 787-822), as the position of the patient and the placement of the electrodes is standardized. This issue is essential in our setting where many people are involved in the measurements.

Other field methods, like skinfold measurement described as prone to subjectivity, have been reported with a high interviewer variability (Kiplstein-Grobusch K, Int J Epidemiol 1997, 26: S174-80). We could not find any study comparing the reproducibility of skinfolds vs. BIA.

 To acknowledge the limitation of BIA, we deleted the following sentence in the last paragraph of the discussion: “The body composition was measured by BIA, using a formula developed in the Geneva population and validated against dual-energy x-ray absorptiometry [16]. We added in the  same paragraph, but under the limitations: “The body composition was measured by several BIA devices whose agreement may be questioned. However, all devices had been cross-calibrated with the same calibration jig and we  used a single formula, developed in the Geneva population and validated against dual-energy x-ray absorptiometry, to derive fat-free mass[16]“.

  12.Page 4 Line 18/19: ‘the lower was the running speed’ can be amended to ‘the lower the running speed’

  This was changed as suggested.

 Discussion

12. ‘…men increased as FMI, and to a lesser extent, body mass index.’ this is an incomplete sentence currently. Please complete this sentence.

 Thank you for the careful reviewing. The sentence was completed and reads now as follows: “This longitudinal cohort study showed that running speed over 5 km in women and 7 km in men increased as FMI, and to a lesser extent, body mass index decreased”.

 13. ‘submitted to a seven-week high-intensity interval training thrice weekly’ may read better as ‘men and women who undertook seven-week high-intensity interval training programme, performed three times per week, decreased…’

 The change was performed as suggested.

 14. As above ‘three times per week’ may read better than ‘thrice’. Whilst this is grammatically correct, the term is not necessarily in common parlance and may be difficult for those whom English is not their primary language.

 As it is grammatically correct, we decided to stay with the term “thrice”

 15. Please provide a reference or two to support your hypothesis that men may lose more fat free mass than women and improve their running speed. This is really interesting, and I think you’re correct in moving towards this hypothesis, it can be developed quite nicely with some supporting references.

 The running performance for a given % muscle mass has been reported to be better in men than women (Perez-Gomez et al, Eur J Appl Physiol 2008; 102: 685-94). Other authors reported a higher anaerobic power reserve (Panissa et al, J Sports Sci Med 2016; 15:372-8) or a higher capacity to perform sprints of long duration in men than women (Mageean et al, Int J Exerc Sci 2011; 4: 229-237). These results suggest that men have a muscle mass which is more efficient metabolically than women, and that they need less muscle mass than women for a given effort.

 We have thus added the following comment in the discussion (end of the second paragraph): “In line with this hypothesis, men were reported to have a higher anaerobic power reserve than women {Panissa, 2016 #802}. This result suggests that men have a muscle mass which is more efficient metabolically than women, and that they need less muscle mass than women for a given effort.”  

16. ‘Cerutti et al, who showed’ can be amended to ‘Cerutti et al, showed’ or consider the word reported here

 Thank you for correcting this error. We changed it for “Cerutti et al. showed”

  17. ‘could maintain longer their running speed…’ should be changed to something that re-emphasises the alternative i.e. sedentary lifestyle e.g. ‘could maintain their running speed longer over the years than their sedentary counterparts.’

 The change was performed as suggested.

 18. ‘The main limitations is’ should read ‘the main limitations are…’

 Thank you for correcting this error. It was changed in the text.

 19. Where you mention drugs, you may wish to clarify that these are pharmaceutical as opposed to recreational.

 We added the term “pharmaceutical”

 20. Your final sentence is good, but please can you clarify the level of runners you are assessing. This will make for a stronger closing statement e.g. ‘but this bias should not influence the relationship between changes in body composition and running speed in recreational runners.’

 Thank you for this good comment, with which we agree. The term “in recreational runners” was added to the final sentence

21. You mention gender specific effects in your discussion, and your results highlight these nicely. Please refer to them in your conclusion and be specific as to which variables negatively affect running performance in which sex. This will make for a more thorough conclusion.

 The first sentence of the conclusion was changed to: “A decrease of FMI in both sexes and a decrease of FFMI in men predict an improvement of running speed over 5 to 7 km”. The second sentence (“The decline of running performance with aging occurs from 50 years onward “),  reads now as: “ “The decline of running performance with aging occurs from 50 years onward in both sexes”.

 22. Consider changing ‘following body composition in ambitious runners…’ to ‘assessing body composition in recreational runners…’

The body composition measurement is especially important for people who want to improve performance-related fitness and thus is related with ambition.

 We suggest changing the term “following body composition in ambitious runners…” to “following body composition in ambitious recreational runners…”.

 23. Supplemental figure 1: Please check that all numbers are correct – there seems to be a large discrepancy in the exclusion figures?

 You are correct and we are sorry for this confusion, which is due to the fact that sometimes we have considered the number of subjects and sometimes the number of observations (BIA + running time).

 In order to avoid any confusion in figure 1, we have focused on the number of observations, which corresponds to the number of BIA measurements associated with a running time. Only in the last box, we have detailed the number of subjects to which this multiple measurements refer, and which are line with the first sentence of the results. We avoided the term “N=” and preferred to detail the numbers by the terms “observations” or “participants”

 24. Supplemental figures are excellent and support your conclusions well – if possible it may be worth including them, and developing the discussion accordingly as this would bring more depth to the data presented/interpreted.

 Thank you for this very nice comment. Although we agree that these figures are useful, we prefer to keep the manuscript short. As such, we hope that it motivates readers and that it allows an easy and fast understanding of the main issues. For those who are interested, they can then rely on the supplemental material.

Round  2

Reviewer 1 Report

Appreciate authors for the responses and revision. 

It is hard to find the revisions in the manuscript, as authors did not mentioned clearly in the responses OR not highlighted in the revised manuscript. 

Author Response

Thank you for your comments. We have uploaded a manuscript with track -change.

Reviewer 2 Report

Dear authors,

Many thanks for the changes made following the first round of revision, it is clear that these have been extensive and the manuscript has improved because of this. I thank you for an informative and interesting response to my previous comments.

Author Response

Thank you for your previous comments and thank you for your positive response.